# Metagenomic Binning Revealed Microbial Shifts in Anaerobic Degradation of Phenol with Hydrochar and Pyrochar

Tao Luo [1,2,3], Jun He [1,2,3], Zhijian Shi [1,2,3], Yan Shi [1,2,3], Shicheng Zhang [1,2,3,4], Yan Liu [1,*] and Gang Luo [1,2,3,4,*]

1   Department of Environment Science and Engineering, Fudan University, Shanghai 200433, China
2   Shanghai Key Laboratory of Atmospheric Particle Pollution and Prevention (LAP3), Department of Environmental Science and Engineering, Fudan University, Shanghai 200433, China
3   Shanghai Technical Service Platform for Pollution Control and Resource Utilization of Organic Wastes, Shanghai 200438, China
4   Shanghai Institute of Pollution Control and Ecological Security, Shanghai 200092, China
*   Correspondence: liuyan@fudan.edu.cn (Y.L.); gangl@fudan.edu.cn (G.L.)

**Abstract:** Phenolic compounds, which are difficultly degraded, are one of the main toxic threats faced in the anaerobic digestion (AD) process. It has previously been reported that hydrochar/pyrochar produced by the hydrothermal liquefaction/pyrolysis of biomass can enhance AD by promoting direct interspecific electron transfer (DIET). The present study investigated the effects of different hydrochars and pyrochars on the anaerobic degradation of phenol and provided deep insights into the related micro-organisms at the species level through genome-centric metagenomic analysis. Compared with the control experiment, the addition of hydrochar and pyrochar shortened the lag time. However, hydrochar created a large increase in the maximum methane production rate ($R_m$) (79.1%) compared to the control experiments, while the addition of pyrochar decreased $R_m$. Metagenomic analysis showed that the addition of carbon materials affected the relative abundance of genes in the phenol anaerobic degradation pathway, as well as the species and relative abundance of phenol degrading micro-organisms. The relative abundance of key genes for phenol degradation, such as bsdB, bamB, oah, etc., under the action of hydrochar was higher than those under the action of pyrochar. In addition, hydrochar-enriched phenol degradation-related bacteria (*Syntrophus aciditrophicus*, etc.) and methanogen (*Methanothrix soehngenii*, etc.). These micro-organisms might improve the phenol degradation efficiency by promoting DIET. Therefore, hydrochar had a more significant effect in promoting anaerobic degradation of phenol.

**Keywords:** anaerobic digestion; hydrochar; pyrochar; phenol; metagenomic analysis





## 1. Introduction

Anaerobic digestion (AD) is an environmentally friendly, sustainable, and economic biological process, and it is widely used in the treatment of organic wastes [1]. AD degrades organic wastes into valuable industrial raw materials, such as methane, volatile fatty acids (VFAs), and alcohol, through the physiological metabolism of micro-organisms, and solves the problems of waste treatment and resource recovery. However, the AD process is still vulnerable to various types of inhibitory compounds (such as ammonia nitrogen, phenols, long-chain fatty acids, etc.), resulting in poor stability and difficulty in dealing with highly toxic substances [2]. Therefore, extensive research has been carried out to explore how to promote the efficiency of AD.

Phenol is a toxic and carcinogenic aromatic compound, which is one of the main threats faced by the AD process. Phenol is a major petrochemical intermediate and is often used as a disinfectant and a reagent in chemical analysis [3–5]. Its biggest industrial use is the production of phenolic resin, such as phenol–formaldehyde resins. Phenol is also used as a solvent in glue, as an extraction solvent in refineries and lubricant production, and as an internal antiseptic and gastric anesthetic in veterinary medicine. Additionally,

phenol is the cornerstone of some synthetic drugs, such as aspirin. Phenol is also present in benzole and coal tar produced during coal coking. It can also be produced through the degradation of natural aromatic substances, such as humic acid and tannin compounds [6]. Previous studies have detected that the phenol concentration in the AD sludge treating various organic wastes can reach 4288 mg/kg [7]. It has been reported that phenol with a concentration of over 250 mg/L can inhibit AD [2,8]. Additionally, the AD of phenol is a complex and slow process, with a thermodynamic barrier (Equations (1)–(4)), leading to its occurrence only through syntrophic metabolism [9–11]. Therefore, it is difficult to completely degraded phenol in the process of AD.

$$C_6H_6O + HCO_3^- + H_2 \rightarrow C_6H_5COO^- + 2H_2O \qquad \Delta G^{0\prime} = -88.4 KJ/mol \qquad (1)$$

$$C_6H_5COO^- + 7H_2O \rightarrow 3CH_3COO^- + HCO_3^- + 3H^+ + 3H_2 \quad \Delta G^{0\prime} = +94.1 KJ/mol \qquad (2)$$

$$CH_3COO^- + H_2O \rightarrow HCO_3^- + CH_4 \qquad \Delta G^{0\prime} = -31.0 KJ/mol \qquad (3)$$

$$HCO_3^- + 4H_2 + H^+ \rightarrow CH_4 + 3H_2O \qquad \Delta G^{0\prime} = -135.6 KJ/mol \qquad (4)$$

Discharged phenol that is not completely degraded would have an adverse impact on the environment and human health [12]. Phenol has germicidal activity associated with its protein denaturing ability [5]. Phenol can be quickly absorbed through the skin and can cause burns to the skin and eyes upon contact with the human body. Excessive exposure may lead to coma, convulsions, cyanosis, and death. Internally, phenol can affect the liver, kidneys, lungs, and vascular system. The ingestion of 1 g of phenol is fatal to humans [13]. Therefore, it is necessary to study how to promote the AD of phenol.

Studies have found that carbon materials such as pyrochar can alleviate the inhibition of phenol and enhance AD, and this can be achieved through two mechanisms: by reducing the stress of phenol through adsorption to alleviate the inhibition, and by enhancing the AD ability of micro-organisms by promoting direct interspecific electron transfer (DIET) [14–17]. In DIET, electrons flow directly from exogenic bacteria to electrogenic methanogens through cell components, such as cytochrome C, or conductive materials, making DIET a more effective method for VFA degradation compared to interspecies hydrogen transfer [18]. Previous studies found that the addition of pyrochars derived from rice straw and manure accelerated methanogenesis remarkably, showing a 10.7 to 12.3 times higher methane production rate compared to the control. In addition, these pyrochars stimulate methanogenesis by facilitating DIET between methanogens and *Geobacteraceae* [19]. Hydrochar is the solid product of biomass hydrothermal liquefaction (HTL), which degrades solid biopolymer structures in the water environment through high temperature and high pressure (200–350 °C, 4–22 Mpa), so as to convert biomass into liquid and solid fuels [20,21]. Compared with pyrolysis, HTL is more suitable for the treatment of biomass with a high moisture content. As the product of HTL, hydrochar has a higher quality and energy density, and better dehydration than raw biomass [22]. It has been found that hydrochar has the ability to improve AD efficiency [23]. Previous studies have shown that sludge hydrochar can increase the methane production rate by 37% at a high organic loading, which may be due to the enhancement of microbial methanogenesis through the promotion of DIET [23]. It is possible that hydrochar can also promote the AD of phenol; however, there is still a lack of research. Reaction temperature is a key parameter in the thermochemical process. Higher temperatures significantly increase carbon content, surface area, and pore volume, while reducing biochar production and oxygen content [24,25]. Therefore, in order to comprehensively compare the effects of pyrochar and hydrochar on phenol AD, different reaction temperatures were included in the experimental design of this study.

The changes of microbial community in the AD process of phenol deserve further study. The degradation of aromatic compounds has been found in nitrate, sulfate, and iron reducing bacteria, including *Rhodopseudomonas palustris*, *Magnetospirillum* spp., *Syntrophus*

*aciditrophicus* etc. [26–29]. In recent years, the research on phenol anaerobic degradation has found several phyla that exist in almost all reactors, namely, *Chloroflexi*, *Proteobateria*, *Firmicutes*, and *Bacteroidetes* [4,26,30–34]. Although phenol degraders have been studied to some extent, our understanding of genetics, microbial physiology, and community ecology is still limited, because more than 99% of environmental prokaryotes cannot be cultured in a laboratory [35,36]. Additionally, phenol degraders at the species level still need further research. Genome-centric metagenomic analysis allows researchers to complete the identification of individual species from a complex environment, and the extraction and observation of their genomic structure, which will promote the discovery of new species and the exploration of their phenol degradation function [36,37].

The purpose of this study is to analyze and compare the effects of hydrochar and pyrochar on the AD of phenol, identify the key microbial species related to the AD of phenol by genome-centric metagenomics, and clarify the changes of microbial communities induced by different carbon materials, in order to provide new knowledge about the AD of phenol.

## 2. Materials and Methods

### 2.1. Substrate and Inoculum

Phenol was directly purchased from Aladdin Chemistry Co., Ltd. Shanghai, China, and used as a substrate. The inoculum was obtained from an AD reactor in an ethanol plant treating cassava stillage (Taicang, Suzhou, China). Its characteristics were TS 37.2 $\pm$ 1.5 g/L, VS 30.1 $\pm$ 1.1 g/L and pH 7.2 $\pm$ 0.1.

### 2.2. Hydrochar and Pyrochar Preparatio

Two kinds of hydrochar and two kinds of pyrochar were prepared from corn straw by hydrothermal liquefaction (HTL) and pyrolysis, respectively. HTL was conducted in a 1 L reactor (E-1000, EadyChem, Beijing, China) at 260 °C for 0.5 h and 320 °C for 0.5 h, which is a common procedure for HTL [20]. The hydrochar was then washed with tetrahydrofuran to remove bio-oil; a detailed description was provided in our previous study [23]. The pyrolysis was carried out in a tubular furnace at 500 °C for 1 h and 700 °C for 1 h. All the carbons were ground and passed through a 3 mm sieve before usage, and they were named H260 (Hydrochar obtained at 260 °C), H320 (Hydrochar obtained at 320 °C), P500 (Pyrochar obtained at 500 °C), and P700 (Pyrochar obtained at 700 °C).

### 2.3. Experimental Set-Up

Batch experiments were conducted to determine the phenol degradation with H260, H320, P500, and P700. For the batch experiments, 118 mL serum bottles were used. The required amounts of phenol, water, carbon material, and BA medium, and 3 mL of inoculum, were added to each bottle to achieve the final working volume of 60 mL, and the carbon material concentration was kept to 10 g/L for each type. The composition of the BA medium was described in a previous study [38]. The concentration of phenol was controlled at 1 g/L, which was similar to that of industrial coal gasification wastewater [16]. A total of 5 g/L of $NaHCO_3$ was added as a pH buffer. The bottles without carbon material were also used as a control (C), and the bottles that contained only inoculum were used as a blank. Nitrogen was used to flush all bottles for 5 min, and the bottles were sealed with butyl rubber plugs and aluminum screw caps. The bottles were placed in an incubator with a constant temperature of 37 °C. All the experiments were performed in triplicate.

### 2.4. Metagenomic Analysis

The samples obtained from anaerobic experiments with H260, H320, P500, and P700 were used for a metagenomic analysis, in order to identify the key species in the AD of phenol, and to understand their shifts with different carbon materials. A PowerMax Soil DNA Isolation Kit (MoBio Laboratories, Carlsbad, CA, USA) was used to extract total genomic DNA, and agarose gel electrophoresis and a Nanodrop ND-2000c (Thermo Fisher Scientific,

Waltham, MA, USA) were subsequently used for quality-checks. The fragmentation of DNA was performed through Covaris M220 (Gene Company, Hongkong, China), and fragments of approximately 400 bp were screened. A PE library was built using the NEXTFLEX Rapid DNA-Seq library kit (Bioo Scientific, Austin, TX, USA). Metagenomic sequencing was then performed after bridge PCR amplification. Metagenome libraries ($2 \times 150$ bp) were sequenced on the Illumina Hiseq 2500 platform (Illumina Inc., San Diego, CA, USA). The raw sequences have been submitted and deposited under BioProject PRJNA915481 on the National Center for Biotechnology Information (NCBI) website. Trimmomatic software (v0.38) was used to filter and trim the paired-end reads with these parameters: LEADING:3, TRAILING:3, SLIDINGWINDOW:4:15, and MINLEN:36 [39]. MEGAHIT (v1.0) was used to carry out the metagenome assembly process to generate one co-assembly. Metagenomic binning was then applied to the co-assembly rely on MetaBAT2, CONCOCT, and MaxBin2. The final GBs were then formed by using MetaWRAP to consolidate the GBs generated from the binning software into a bin set. The characteristics of the GBs were determined by CheckM (v1.0.18). In addition, Prokka (v1.13.7) was used to annotate the GBs and CAT (v5.12). GBDK-Tk (v1.3.0) was used to perform taxonomic classifications assigned to GBs. The relative abundance of genes was calculated on the basis of gene recognition data processed by Prodigal (v2.6.3). GhostKOALA was used to measure the metabolic mechanism and functional analysis of GBs. The coding sequences (CDS) dataset of GBs were annotated with the Kyoto Encyclopedia of Genes and Genomes (KEGG) and the Evolutionary Genealogy of Genes: Non-supervised Orthologous Groups (eggNOG). A Pearson analysis between $R_m$ and GBs was calculated using SPSS 20.0.

### 2.5. Analytical Methods

The methane content in the headspace of the bottles was measured by gas chromatography (GC) equipped with a thermal conductivity detector (GC-960, Haixin, China), with $N_2$ as the carrier gas. The temperatures of the injector, detector, and oven were 190 °C, 110 °C, and 190 °C, respectively [40]. The concentrations of SCFAs, MCFAs, ethanol, and phenol were measured by GC (GC 2010 Plus, Shimadzu, Kyoto, Japan) equipped with a flame ionization detector, in accordance with a previous study [41], and the chromatographic column was HP-FFAP (Agilent, Santa Clara, CA, USA), with a specification of $30 \text{ m} \times 0.25 \text{ mm} \times 0.25 \text{ μm}$. TS and VS were measured according to Standard Methods (APHA 1995). The analysis of variance (ANOVA) calculated in Excel was used to test the significance of the results.

The adsorption test method and the modified Gompertz model are described in the Supporting Materials.

## 3. Results and Discussion

### 3.1. Effects of Different Carbon Materials on Methane Production from Phenol

Figures 1–3 show the changes in cumulative methane production, phenol concentration, and VFA concentration in the experiments with and without carbon materials, and Table 1 shows the kinetic parameters of methane production.

**Table 1.** Summary of the kinetic parameters of methane production by carbon addition.

| | Measured Methane Production at the End of Batch Experiments (mL) | P (mL) | $R_m$ (mL/d) | λ (d) | $R^2$ | Increasing Rate of $R_m$ (%) * |
|---|---|---|---|---|---|---|
| C | $49.5 \pm 2.9$ | $54.0 \pm 7.1$ | $2.49 \pm 0.41$ | $74.2 \pm 1.5$ | 0.951 | |
| H260 | $51.3 \pm 1.8$ | $50.3 \pm 1.0$ | $4.40 \pm 0.51$ | $59.4 \pm 0.7$ | 0.988 | 76.7 |
| H320 | $48.2 \pm 1.3$ | $49.4 \pm 0.7$ | $4.46 \pm 0.42$ | $57.6 \pm 0.5$ | 0.993 | 79.1 |
| P500 | $50.2 \pm 0.2$ | $51.0 \pm 6.0$ | $2.07 \pm 0.32$ | $66.8 \pm 1.6$ | 0.960 | −16.9 |
| P700 | $46.1 \pm 1.7$ | $54.0 \pm 9.2$ | $1.59 \pm 0.21$ | $63.2 \pm 1.8$ | 0.952 | −36.1 |

\* The increase of $R_m$ was significant ($p < 0.05$).

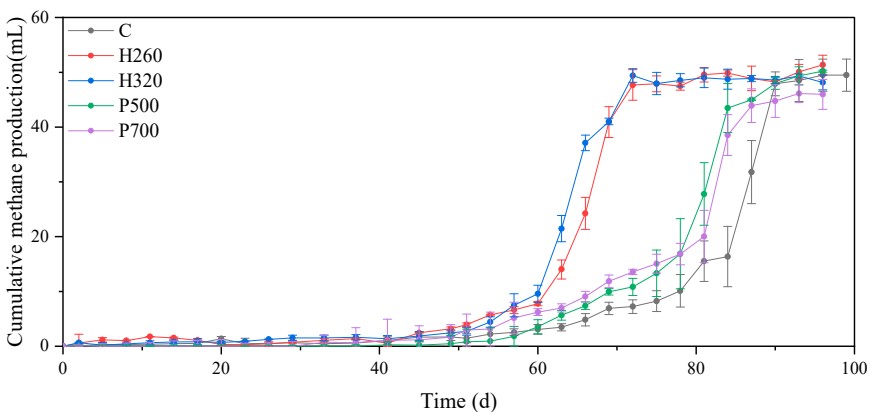

**Figure 1.** Changes of cumulative methane production after adding H260, H320, P500, and P700.

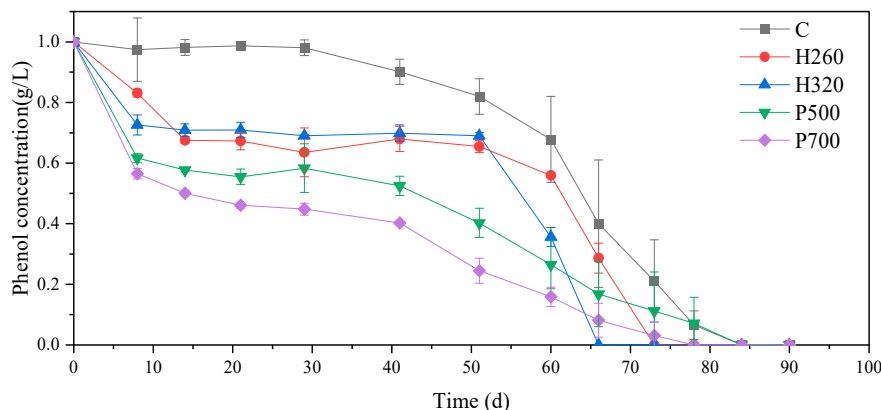

**Figure 2.** Changes of phenol concentration after adding H260, H320, P500, and P700.

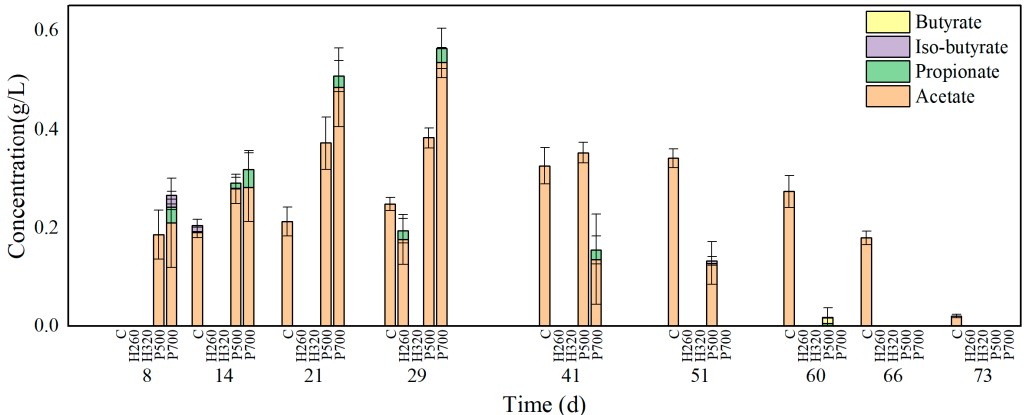

**Figure 3.** Changes of VFA concentration after adding H260, H320, P500, and P700.

There was no significant difference in the maximum methane production across all experimental groups ($p > 0.05$). The four carbon materials obviously shortened the lag time, and the hydrochar group shortened the lag time more. The lag time of the C was 74.2 d, while H260 and H320 shorten the lag time to 59.4 d and 57.6 d, respectively. P500 and P700 shorten the lag time to 66.8 d and 63.2 d, respectively. Previous studies found that the lag time of phenol anaerobic degradation was linearly related to the adsorption capacity of biochar [10]. The adsorption capacity of the carbon materials in this study for phenol is shown in Table S1. It can be seen that the higher the adsorption capacity of phenol by hydrochar and pyrochar prepared at different temperatures, the shorter the lag time; however, the linear relationship between the adsorption capacity and the lag time was not

significant among the different carbon materials, which indicated that there were other potential factors affecting the lag time in addition to the adsorption capacity. Hydrochar also performed better in promoting the maximum methane production rate. Compared with the C, H260 and H320 increased $R_m$ by 76.7% and 79.1%, respectively. Previous studies have found that the surface oxygen-containing functional groups of hydrochar improved anaerobic efficiency by promoting DIET [23]. However, the $R_m$ of P500 and P700 groups decreased by 16.9% and 36.1%, respectively. Pan et al. found that apricot shell-based pyrochar completely inhibited methanogenesis in the AD of phenol, and put forward an inhibitory mechanism: the exposure of absorbed microbes on the shell-based pyrochar to the highly concentrated phenol in the pyrochar's pores resulted in the inhibition of methanogens, especially for *Methanosarcina* [42]. This mechanism might also occur in the pyrochar groups of this experiment.

Before the end of the lag time, the phenol concentration in the group with carbon materials was 29.6–54.3% lower than in the C, which might be mainly due to the adsorption of phenol on carbon materials. The phenol concentration of the pyrochar group was lower. The adsorption capacity of the carbon materials for phenol in this study is shown in Table S1, and was consistent with previous results that found that phenolic compounds' adsorption by pyrochar was superior to their adsorption by hydrochar [43]. However, the detection of VFA also indicated that a small amount of phenol might be degraded to VFA during the lag time. After 65 days, the phenol concentration of the hydrochar groups decreased to 0 before that of the pyrochar groups, indicating that the overall phenol degradation of the hydrochar groups was faster, which was due to the shorter lag time and higher $R_m$ of the hydrochar groups

As shown in Figure 3, the cumulative VFA was mainly acetate. In all the experiments, the concentration of VFA remained at a low level, and the highest value (P700, 29d) was also lower than 0.6 g/L, which was mainly detected before the end of the lag time. In the hydrochar groups, VFA was rarely detected. VFA was only detected in H260, with concentration lower than 0.2 g/L on the 29 d, while the H320 group did not detect VFA at all. After the end of the lag time, the accumulated acetate was rapidly consumed, and no VFA was detected later, which indicated that the consumption rate of acetate was higher than that of the acetate production caused by phenol degradation

The above results show that the phenol degradation and methanogenesis were more active, the methane production rate was higher, and the lag time was shorter in the hydrochar groups. Previous research on the promotion of AD by carbon materials found that it was neither electrical conductivity nor the total redox property of hydrochars and AC, but the abundances of surface oxygen-containing functional groups that correlated to the methane production rates [23]. In addition, studies have shown that hydrochar has more surface oxygen-containing functional groups because water participation in HTL contributes to the formation of oxygen-containing groups [22,43,44]. This may also be the reason why hydrochars had a better promoting effect on AD than pyrochars in this study. Although most pyrochar has been shown to promote phenol degradation before [10,17,42], it had no effect on $R_m$ in this study. Previous studies have also found that pyrochar might inhibit methanogenesis in phenol environments [42], which might be related to the different types of pyrochar used in different studies. However, in this study, the same raw material was used to prepare different pyrochars and hydrochars, and the overall effect of hydrochar was better.

*3.2. Metagenomic Analysis*

3.2.1. Genes Involved in Phenol Anaerobic Degradation

The functional genes were annotated using the KEGG database in order to identify the relative abundance of phenol anaerobic degradation genes in the whole metagenome (Table S2). All the relative abundances were expressed as "genome copies per million reads", which should already be standardized to the individual sample size. The results

showed that the relative abundance of phenol anaerobic degradation genes in the samples under different conditions was significantly different.

The anaerobic biodegradation of phenol can be divided into three stages. The first stage is the reaction of the group on the benzene ring. Phenol is carboxylated to 4-hydroxybenzoate (4OHBz) by 4-hydroxybenzoic acid decarboxylase in the para position, or first converted to phenyl phosphate by phenyl phosphate synthase, and then phenyl phosphate carboxylated to 4OHBz by phenyl phosphate carboxylase. 4OHBz can undergo thioesters with CoA, transforming into 4OHBz-CoA. Then 4OHBz-CoA was reductively dehydroxylated to benzoyl-CoA by 4OHBz-CoA reductase. HbaA (4-hydroxybenzoate-CoA ligase) and hcrABC (4-hydroxybenzoyl-CoA reductase) that can convert 4OHBz-CoA into benzoyl-CoA were not detected in any of the microbial samples. Previous studies found that *Rhodopseudomonas palustris* with cassette disruption of the hcrC gene could not grow anaerobically on 4OHBz, which indicated that hcrABC is necessary for phenol anaerobic degradation [45], and also means that unknown genes expressing hbaA and hcrABC isoenzymes might exist in the microbial samples of this experiment. Although the bsdB of 4-hydroxybenzoate decarboxylase (bsdBCD) had a high relatively abundance and existed in most of the micro-organisms in this study, the peptide corresponding to bsdB also worked as flavopenyltransferase, which plays a role in the biosynthesis of ubiquinone and C5 isoprene. It was found that bsdBCD must have all subunits in order to play a full catalytic and reversible decarboxylation/carboxylation function [46]. Therefore, micro-organisms with bsdC and bsdD genes can clearly be recognized to have the ability to carboxylate phenol. However the relative abundances of these two genes were low in all samples (bsdC < 25, bsdD < 1.3). In addition, micro-organisms with only bsdB genes may also have the potential to carboxylate phenol. The bsdB gene was enriched in the hydrochar group; P500 compared with the C, whereas it was not enriched in P700, indicating that the phenol carboxylation potential in the hydrochar group and P500 was higher. Therefore, the hydrochar probably improved the phenol carboxylation potential by enriching the bsdB gene.

The second stage was the AD of benzoyl-CoA. Benzoyl is regarded as the central intermediate in the AD of various aromatic compounds. Benzoyl-CoA was first dearomatized by benzoyl-CoA reductases: bcrABCD for facultative anaerobes, and bamBC for obligate anaerobes. After dearomatization, cyclohexa-1,5-diene-1-carbonyl-CoA can take two routes (bam and bad routes) to the cleave of the aromatic ring. In the bam route, cyclohexa-1,5-diene-1-carbonyl-CoA is hydrated to 6-hydroxycyclohex-1-ene-1-carbonyl-CoA and then reduced to 6-oxocyclohex-1-carbonyl-CoA. Then the ring is cleaved to 3-hydroxypimelyl-CoA by a hydrolase. In the bad route, cyclohex-1-ene-1-carboxyl-CoA is hydrated and reduced to 2-ketocyclohexane-1-carboxyl-CoA. The ring is then hydrolyzed to yield pimeloyl-CoA, which is subsequently still reduced and hydrated to 3-hydroxypimelyl-CoA. The 3-hydroxypimelyl-CoA then continues to be degraded to acetyl-CoA. In this stage, the genes of two kinds of benzoyl-CoA reductase (bcrABCD, bamBC) were detected, but as shown in Table S2, the relative abundance of bcrABCD was much lower, and therefore bamBC was the main benzoyl-CoA reductase. The reason may be that bcrABCD mainly exists in facultative anaerobes, which generally have no competitive advantage in strict anaerobic environments [47]. The ring opening pathway after dearomatization was mainly the bam route, but a small number of bad route genes were also detected. In addition, oah genes in the bam route, together with bamB and bcrC genes, were used as an indicator of the ability of anaerobically degraded aromatic compounds [48,49]. In the samples of this study, the C had the highest relative abundance of bamB (354.82) and oah (193.08). This might be because in the absence of carbon materials, phenol had the greatest inhibitory effect on the C, leading to the enrichment of genes directly related to phenol, while indirectly related genes such as aceticlastic methanogenesis, discussed below, were fewer. The relative abundances of bamB and oah in the experimental groups were arranged in the following order: H320 > H260 > P500 > P700, which was similar to the sequence of anaerobic degradation performance, indicating that the relative abundance

of benzoyl-CoA pathway genes was closely related to the efficiency of anaerobic methane production. As shown in Table S2, compared with pyrochar, hydrochar retained a higher relative abundance of benzoyl-CoA pathway genes, which helped the communities with hydrochar have greater efficiency of anaerobic phenol production.

Finally, acetate converted by acetyl-CoA entered the methanogens to participate in the aceticlastic methanogenesis in the third stage. As shown in Table S2, the gene pathway of aceticlastic methanogenesis was completed in all groups, and the relative abundance was the highest in the hydrochar groups, while the abundance in the pyrochar groups was lower than in the hydrochar groups. However, there was no obvious difference. The enzyme that directly reduce acetate/acetyl-CoA is acetyl-CoA decarbonylase/synthase (ACDS) complex. The key enzyme for methanogenesis is methyl coenzyme M reductase. In addition, the hdrDE enzyme is a unique heterodisulfide reductase of aceticlastic methanogenesis, which is different from other methanogenesis pathways. These enzymes might all be responsible for the accumulated acetate being rapidly consumed, but the relative abundance changes of the genes expressing these enzymes were basically the same among different groups.

In general, the above results indicate that the addition of four carbon materials affected the relative abundances of genes relating with phenol anaerobic degradation, which resulted in the different methane production performances. Hydrochar clearly enriched the related genes in the first and third stages, and in the second stage, it retained a higher relative abundance of benzoyl-CoA pathways than pyrochar. These changes in the genes may explain the promotion of hydrochar on phenol degradation.

### 3.2.2. Genome Reconstruction and Metabolic Potential

A total of 208 GBs were obtained after metagenomic analysis, and 206 GBs were of high quality, i.e., integrity $\geq 70\%$ and pollution $\leq 5\%$ (Table S3). Most GBs (>65%) could not be allocated to the species level, suggesting that they had not been characterized previously and may be new phenol anaerobic degrading species (Table S4). The relative abundances of high-quality GBs in different samples are shown in Figure 4. The relative abundances are also expressed as "genome copies per million reads". The functional genes were annotated through the KEGG for the identification of potential GBs capable of anaerobic phenol degradation (Tables S5 and S6). A Pearson analysis identified 14 GBs that were significantly positively correlated with $R_m$ and 32 GBs that were significantly negatively correlated with $R_m$ (Table S7).

Among the GBs definitively containing the bsdBCD gene, *RBG-16-64-13* sp. FDU161 had the highest relative abundance (9.66~15.92). It was noted in the GTDB classification as a species of the *RBG-16-64-13* family. The original *RBG-16-64-13* species were found to have the ability to assimilate formaldehyde in a genome study on acquifer fractions and groundwater microbial communities [50]. In our study, *RBG-16-64-13* sp. FDU161 was only found to have an incomplete formaldehyde assimilation pathway through ribose monophosphate after gene annotation, but it contained genes that could metabolize glucose to acetic acid. In addition, *DTU098 sp002305915* FDU100 was more common in the C and the two pyrochar groups. This genome was sequenced in previous anaerobic digester metagenome studies [51]. *UBA4789* sp. FDU132 only existed in H320, and the *UBA4789* genus was merged into the *Syntrophopionicum* genus. *DTU098 sp002305915* FDU100 and *UBA4789* sp. FDU132 both come from the *Pelotomalaceae* family, whose members are mainly propionate oxidizing bacteria, which may grow with methanogens syntrophically in anaerobic systems [52]. Additionally, *Cryptanaerobacter phenolicus*, a close relative of the genus *Pelotomaculum*, is one of the few known species that can transform phenol and 4-OHBz into benzoate [53]. The family *Pelotomalaceae* may have the potential for phenol carboxylation. The gene annotation of *DTU098 sp002305915* FDU100 only found an incomplete propionate oxidation pathway. Incomplete formaldehyde assimilation and a sulfate reduction pathway were also found. *UBA4789* sp. FDU132 also had an incomplete propionate oxidation pathway and an incomplete formaldehyde assimilation pathway. The total relative abundance of determined bsdBCD containing micro-organisms in different

experimental groups was too low, and the reaction on the benzene ring might be mainly completed by other potential micro-organisms.

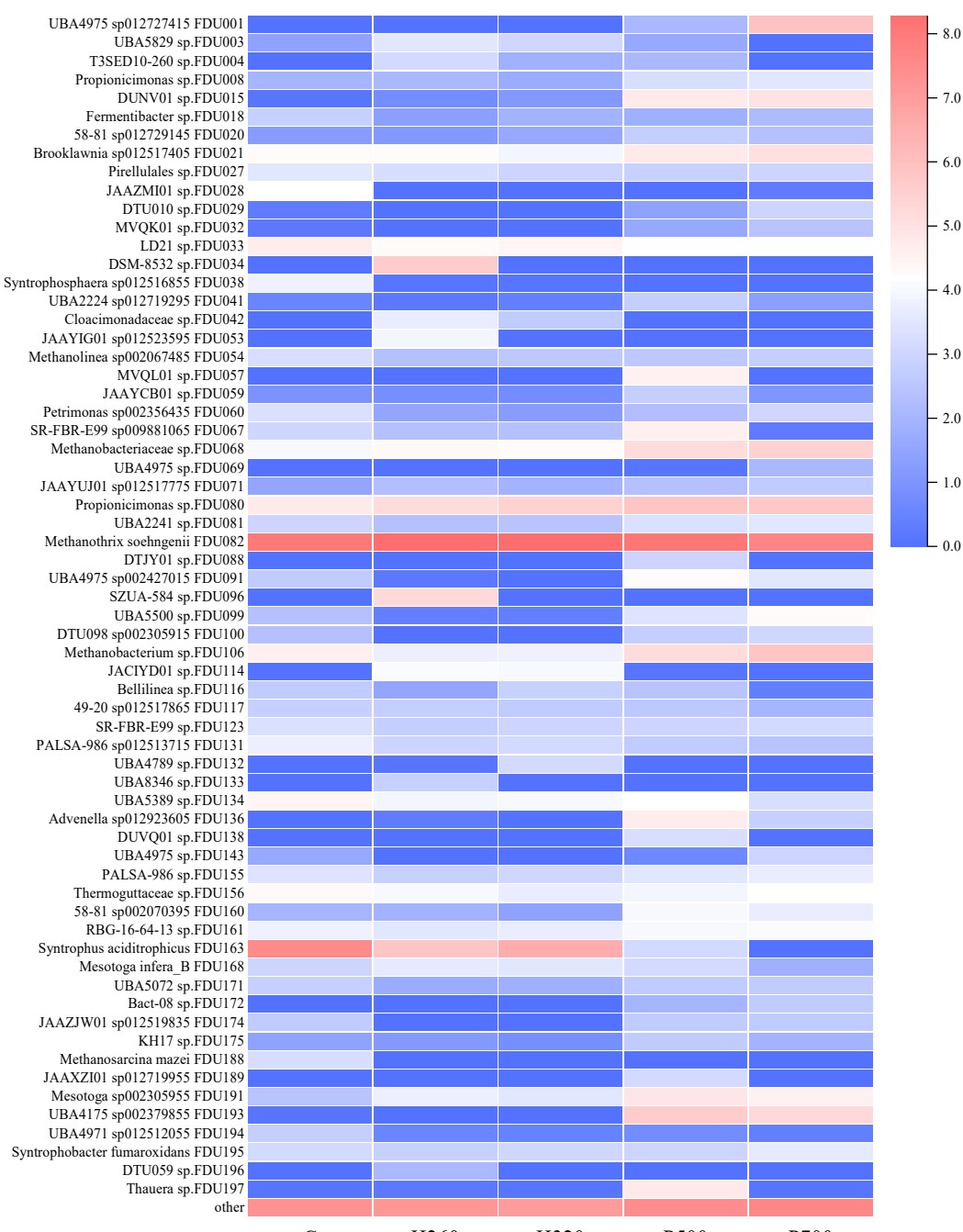

**Figure 4.** The abundances of GBs in all the samples. "Others" includes all the GBs that have abundances less than 5. The values in the legend were the log2 transformation of abundances.

The species and relative abundance of micro-organisms involved in the degradation of benzoyl-CoA varied greatly among different samples. In the C, the micro-organisms involved in the ring opening of benzoyl-CoA were mainly *Syntrophus acidotrophicus* FDU163 with a high relative abundance (179.7) and *JAAZMI01* sp. FDU028 with a low relative abundance (17.4). *S. aciditrophicus* is a strictly anaerobic, gram-negative, non-motile, non-spore forming rod-shaped bacteria, which can degrade benzoate and some fatty acids through the syntrophic growth with hydrogen/formate-consuming micro-organisms [54–56]. Previous studies found that *S. aciditrophicus* can produce conductive pili (e-pili), which is

necessary for DIET and can grow through DIET [57], and it showed the possibility of DIET during phenol degradation in this experiment. Hydrochar might promote phenol degradation efficiency by promoting DIET between *Syntrophus aciditrophicus* FDU163 and its syntrophic methanogens. The gene annotation of *JAAZMI01* sp. FDU028 showed that in addition to the benzoic acid metabolic pathway, it also had an Embden–Meyerhof pathway for glycolytic, acetyl-CoA pathway and parts of ribulose monophosphate pathway for formaldehyde assimilation, which meant that although *JAAZMI01* sp. FDU028 would compete with *S. aciditrophicus* in benzoic acid metabolism, it could also grow syntrophically with *S. aciditrophicus* as a consumer of hydrogen/C1 compounds. In two hydrochar groups, *S. aciditrophicus* was also the main ring opener, but its relative abundance decreased to 56.2 and 94.3, respectively. *JACIYD01* sp. FDU114 was another ring opener in the hydrochar group, and its relative abundance in the two hydrochar groups was close (16.3 and 15.3). *JACIYD01* sp. FDU114 was annotated to the *JACIYD01* genus, which was once found in the thermophilic anaerobic methanogenic environment for paraffin degradation [58]. The gene annotation of *JACIYD01* sp. FDU114 showed that it had glycolysis and gluconeogenesis pathways, as well as the ribose monophosphate pathway for formaldehyde assimilation. In P500, the relative abundance of *S. aciditrophicus* was 7.4, which were replaced by *DUNV01* sp. FDU015 (26.6) and *MVQL01* sp. FDU057 (22.8) as the main ring openers. In P700, the ring opener was *DUNV01* sp. FDU015 (31.5) with the disappearance of *S. aciditrophicus*. *DUNV01 sp016841605* (named *Dethioactor alkaliphilus* in NCBI classification), which came under the *DUNV01* genus was a confirmed benzoate degrader [59]. *DUNV01* sp. FDU015 had been annotated and had its benzoate degradation ability confirmed. Its glycolysis and gluconeogenesis pathways were incomplete, but it could fix carbon element through the crassulacean acid metabolism and phosphate acetyltransferase–acetate kinase pathway. *MVQL01* sp. FDU057 was annotated to the *MVQL01* genus. *MVQL01 sp002067205* of this genus had the ability of 4OHBZ metabolism, and it was also found that it encoded benzoate production and two types of electron-confurcating hydrogenases, but did not express benzoate production and used the trimeric hydrogenase [60]. After KEGG annotation, it was verified that *MVQL01* sp. FDU057 had the ability to degrade benzoic acid. It had the pathway of gluconeogenesis to fructose, but the metabolic pathway of glucose was incomplete, and it also had an incomplete pathway of formaldehyde assimilation. *Syntrophus aciditrophicus* FDU163, as a phenol degrader in the C, occupied an absolutely dominant position, but the addition of carbon materials was not conducive to the enrichment of this GB, and it was not even detected in P700. In the hydrochar groups, the decrease of the relative abundance of *Syntrophus aciditrophicus* FDU163 might be caused by the increase of micro-organism in the other steps. In the pyrochar groups, it was mainly replaced by other benzoic acid degraders, and the relative abundance of the substitute (49.4 in P500, 31.5 in P700) was of a low level, which might be one of the reasons for the slow metabolism of phenol. The inhibition of pyrochar on *Syntrophus aciditrophicus* FDU163 might be due to, as mentioned above, the high concentration of phenol adsorbed on the surface of pyrochar that had a high toxicity to micro-organisms; however, this may need further experimental proof.

The GBs with methanogenic pathways were mainly *Methanobacteriaceae* sp. FDU068, *Methanothrix soehngenii* FDU082, and *Methanobacterium* sp. FDU106. In all samples, *Methanothrix soehngenii* FDU082 had the highest relative abundance, which could be speculated to mean that it is the micro-organism mainly completing the methanogenic process. The relative abundance of *Methanothrix soehngenii* FDU082 in the hydrochar group was approximately 20% higher than in the C, while the relative abundance in the P700 was 21% lower than in the C. It was obvious that hydrochar promoted the enrichment of *Methanothrix soehngenii* FDU082, but pyrochar did not. *Methanothrix soehngenii* performs only aceticlastic methanogenesis, and other methanogenic substrates are not used [61]. The KEGG gene annotation verified that *Methanothrix soehngenii* FDU082 had aceticlastic methanogenesis pathways and acetyl-CoA pathways. As acetyl-CoA is the main anaerobic degradation product of benzoic acid, *Methanothrix soehngenii* FDU082 with aceticlastic methanogenesis became the main and most important species, and its relative abundance was significantly

positively correlated to $R_m$. Therefore, the enrichment of *Methanothrix soehngenii* FDU082 might also explain the promotion of hydrochar on the anaerobic metabolism of phenol. In addition, having reductive acetyl-CoA pathways means that *Methanothrix soehngenii* FDU082 might also produce methane by using $H^+$ and $e^-$ through DIET. *Metanobacteriaceae* sp. FDU068 was annotated to the *Metanobacteriaceae* family, which mainly produces methane through hydrogenotrophic methanogenesis. The annotation showed that apart from the hydrogenotrophic methanogenesis pathway, *Methanobacteriaceae* sp. FDU068 could reduce $CO_2$ through the acetyl-CoA pathway and carried out the incomplete reductive citrate cycle to fix carbon. *Metanobacterium* sp. FDU106 was annotated to the *Metanobacterium* genus under the *Metanobacteriacee* family. The annotation result showed that it also had genes of hydrogenotrophic methanogenesis, acetyl-CoA pathways, and the incomplete reductive citrate cycle. The relative abundances of *Metanobacteriaceae* sp. FDU068 and *Metanobacterium* sp. FDU106 in the C and hydrochar groups were lower than in the pyrochar group. Due to the fact that *Methanobacteriaceae* sp. FDU068 and *Methanobacterium* sp. FDU106 were mainly responsible for hydrogenotrophic methanogenesis, their contribution to phenol degradation was not direct, but they were likely to be involved in the metabolism of intermediates generated by *Syntrophus aciditrophicus* FDU163 or the other micro-organism.

## 4. Conclusions

This study can provide new ideas for enhancing the phenol AD process, and also search for potential application directions for hydrochar. The key microbial species related to phenol AD were identified through genome-centered metagenomics, providing new knowledge about phenol AD micro-organisms. However, further research is needed to explore the deeper microbial mechanisms and the mechanism by which pyrolysis carbon inhibits AD. Metagenomic analysis can only display the potential function of each GB, and further gene expression correlation analysis is needed to clearly determine the active micro-organisms and gene pathways in the phenol AD process. The straw pyrochar in this study showed an inhibitory effect on the maximum methane production rate of phenol AD. In most studies, the pyrochar only promoted phenol AD. Further research is needed on the impact of different pyrochars on phenol AD, in order to gain a more comprehensive understanding of the mechanism of carbon materials' impact on phenol AD.

This study showed that all four carbon materials affected the anaerobic degradation of phenol. They all shortened the lag time of methanogenesis. Hydrochar significantly increased the maximum methane production rate, while pyrochar decreased it. Metagenomic analysis showed that the addition of carbon materials affected the relative abundance of genes in the anaerobic phenol degradation pathway and the relative abundance of phenol degrading micro-organisms. The relative abundance of bsdB, bamB, oah, and other key genes, as well as the relative abundances of phenol degrading micro-organisms and *Methanothrix soehngenii* in the hydrochar group, were higher than those in the pyrochar group, which might be the reason that hydrochar more efficiently promoted phenol degradation. Hydrochar might improve the phenol degradation efficiency of micro-organisms by promoting DIET. Although the adsorption properties of carbon materials were conducive to shortening the lag time, an adsorption capacity that is too high, such as P700, may not be conducive to the enrichment of phenol degrading micro-organisms, and had negative impacts on the efficiency of phenol degradation and methanogenesis.

**Supplementary Materials:** The following supporting information can be downloaded at https://www.mdpi.com/article/10.3390/fermentation9040387/s1: Table S1: Phenol adsorption capacity of carbon materials; Table S2: Summary of relative abundance of related genes; Table S3: Summary of the GBs with high quality; Table S4: Taxonomic classification of the GBs; Table S5: Summary of KEGG annotation of the GBs with high quality; Table S6: Main pathways/genes in AD process of the GBs; Table S7: The Pearson analysis of the correlation between Rm and GBs.

**Author Contributions:** Conceptualization, T.L.; methodology, T.L., J.H. and Z.S.; investigation, T.L. and J.H.; data curation, T.L.; writing—original draft preparation, T.L.; resources, Y.S. and S.Z.; supervision, Y.L. and G.L.; writing—reviewing and editing, G.L.; visualization, G.L. All authors have read and agreed to the published version of the manuscript.

**Funding:** This work was financially supported by the National Natural Science Foundation of China (31970117), and the Science and Technology Commission of Shanghai Municipality (19DZ1204704, 22ZR1405900).

**Institutional Review Board Statement:** Not applicable.

**Informed Consent Statement:** Not applicable.

**Data Availability Statement:** The raw sequences have been submitted and deposited under BioProject PRJNA915481 on the National Center for Biotechnology Information (NCBI) website.

**Conflicts of Interest:** The authors declare no conflict of interest. The funders had no role in the design of this study; in the collection, analysis, or interpretation of data; in the writing of the manuscript; or in the decision to publish the results.

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
