# Peer review of "Metagenomic Binning Revealed Microbial Shifts in Anaerobic Degradation of Phenol with Hydrochar and Pyrochar"

_fermentation, doi:10.3390/fermentation9040387_

Round 1

Reviewer 1 Report

The work is interesting and may be valuable for people interested in degradation of phenol using specified materials. Works aiming in pollutants removal are always interesting and valuable.

Introduction is concise providing main information on the topic.

Methods are not completely described.

There are missing details on PCR and sequences used.

Why GC analysis has been removed from the main text? It is important part of the study.

I miss the name of the software used for conducting statistical analysis.

From Fig. 3 it is hard to see other fraction than acetate. Maybe the graph could be improved by using breaks and making small fractions more visible?

Reviewer 2 Report

1) Write biological names in italics. More than 20 mistakes.

2) REF formatting mistake.

Example =  et al[20-23]

More than 25 such mistakes can be seen.

3) Did the authors really check the PDF when it was uploaded?

4) Why did the authors submit a VERSION which is very carelessly prepared and with very bad formatting?

5) Discuss the role of different microorganisms involved in phenol degradation, their pathway, and Gibbs free energy.

6) Show all the reactions involved for phenol degradation, the the end products. 

7) How was pH change manifested and how did the authors overcome the problem of end product accumulation in the AD system?

8) What are the case-based applications of phenol in gasification, pharmacy, pesticide and oil refining? discuss in detail.

9) Discuss the adverse impact on the environment and human health in one new paragraph.

10) What are the legislations regarding phenol in China, and how does that compare with other Asian countries?

11) How was quality assurance and statistics taken care in the different bottles and the different experiments.

12) Provide model number, city/country, name for all the electrodes, columns, equipments, etc.

13) Discuss the following aspects in MORE DETAIL:

a) Reason for enhanced phenol degradation with the hydrochar groups - compare with literature.

b) What enzymes were responsible for the accumulated acetate being rapidly consumed?

c) State all the real mechanisms by which pyrochar might inhibit methanogenesis?

d) Show examples of the reversible decarboxylation/carboxylation reactions.

e) What are the real reasons for C to have the highest relative abundance of bamB? Discuss them with the results.

f) In which industries are  thermophilic anaerobic methanogenic environment prevailing. Give 5-6 examples and justify the findings.

14) Write the practical applications and future research prospects - in 200 words before the conclusions.

Reviewer 3 Report

In this manuscript, authors tried to provide some insights into the microbial shift in anaerobic degradation of phenol with hydrochar and pyrochar. This study can provide some useful information for the future degradation of phenolic substances by AD process. The manuscript can be published at Fermentation after addressing the following comments.

(1) In the Introduction, authors claimed that AD has been widely used in the treatment of organic wastes and wastewater at present. But AD has not been used for wastewater treatment. It is probably used for treating the sludge from wastewater treatment plants.

(2) Authors need to add relevant information of previous studies about pyrochar used in AD into the introduction.

(3) Authors should provide reasons for selecting two types of hydrochar and two types of pyrochar.

(4) L300: Pelotomaculum should be discussed more. It closely related to Cryptabaerobacter, which is responsible for the degradation of phenol to Benzoyl-CoA Comparative analysis with the genome of Cryptabaerobacter should be done.
